# XPERT: Expert Knowledge Transfer for Effective Training of Language Models

**Chang Liu** [1 2]   **Boyu Shi** [1 2]   **Xu Yang** [1 2]   **Xin Geng** [1 2]

## Abstract

Mixture-of-Experts (MoE) language models organize knowledge into explicitly routed expert modules, making expert-level representations traceable and analyzable. By analyzing expert activation patterns in MoE large language models (LLMs), we find that a subset of experts is consistently activated across diverse knowledge domains. These common experts encode cross-domain, generalizable knowledge that is closely related to model generalization, naturally raising the question of how such identifiable expert knowledge can be practically reused. Motivated by this observation, we propose **XPERT**, a framework that extracts, consolidates, and reuses expert knowledge from pre-trained MoE LLMs to support more effective training of language models across different model scales. XPERT identifies cross-domain experts via inference-only analysis, refines their representations through tensor decomposition, and adapts the extracted knowledge to reuse in downstream models. Experiments on language understanding and dialogue generation benchmarks show that models benefiting from reused expert knowledge achieve consistently stronger performance and faster convergence compared to strong baselines. These results highlight MoE LLMs as structured and reusable knowledge sources, and demonstrate the value of expert-level knowledge reuse for improving model training.

## 1. Introduction

With the rapid advancement of large language models (LLMs) (Wiggins & Tejani, 2022; Chowdhery et al., 2023; Achiam et al., 2023; Grattafiori et al., 2024; Zhong et al., 2025), many models have demonstrated strong performance across a wide range of tasks and domains (Zhang et al., 2024; Sun et al., 2024; Yi et al., 2024). While early LLMs were predominantly based on dense architectures (Touvron et al., 2023; Mann et al., 2020; Bai et al., 2023), recent research has increasingly shifted towards sparse Mixture-of-Experts (MoE) models (Shazeer et al., 2017a; Dai et al., 2024; Muennighoff et al., 2025; Team et al., 2024; Liu et al., 2024), which scale model capacity by activating only a small subset of specialized experts per input. This conditional computation paradigm enables MoE models to achieve strong performance while reducing training and inference costs.

Beyond computational efficiency, a key property of MoE architectures is that knowledge is explicitly modularized at the level of experts. Through routing mechanisms, each expert is selectively activated by specific inputs, making the relationship between expert representations and task characteristics both explicit and traceable. Motivated by this property, we analyze expert activation patterns in MoE LLMs using data from different knowledge domains.

We observe that while many experts are domain-specific, a subset of experts remains consistently active across diverse domains. As shown in Figure 1, certain experts (e.g., experts 8, 17, and 30 in OLMoE-7B) exhibit high activation across multiple domains, indicating that they encode shared cross-domain knowledge. We refer to such experts as *common experts*. The existence of common experts suggests a close connection between expert-level representations and model generalization. This intuition is further supported by recent MoE designs such as DeepSeekMoE (Dai et al., 2024), which explicitly introduce shared experts activated across all inputs, highlighting the importance of general-purpose expert knowledge. These observations naturally raise a key question: ***what practical use can be made of expert knowledge that is both identifiable and closely tied to generalization?***

In the vision domain, prior work has shown that generalizable knowledge extracted from pre-trained convolutional neural networks or vision Transformers can be reused to

[1]School of Computer Science and Engineering, Southeast University, Nanjing, China [2]Key Laboratory of New Generation Artificial Intelligence Technology and Its Interdisciplinary Applications (Southeast University), Ministry of Education, China. Correspondence to: Xu Yang <xuyang_palm@seu.edu.cn>, Xin Geng <xgeng@seu.edu.cn>.

*Proceedings of the 43rd International Conference on Machine Learning*, Seoul, South Korea. PMLR 306, 2026. Copyright 2026 by the author(s).

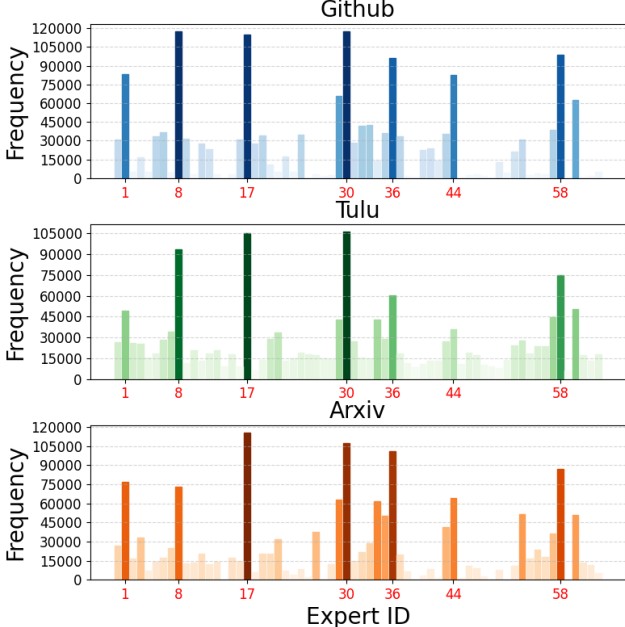

*Figure 1.* Experts activation frequencies of layer 15 in OLMoE-7B across different domains. Additional examples are provided in Appendix A.

improve training efficiency and stability across tasks and model configurations (Wang et al., 2023a; Shi et al., 2024). Inspired by this line of work, we explore whether expert knowledge in MoE-based LLMs can be similarly extracted and transferred to improve the efficiency and effectiveness of model training. We propose **XPERT**, a framework that extracts cross-domain common expert knowledge from MoE LLMs and reuses it to support more effective training of models across different scales.

Under the XPERT framework, expert knowledge is identified through a purely inference-based analysis. As illustrated in Step 1 of Figure 2, we perform forward passes on data drawn from multiple knowledge domains and record expert activation patterns. Experts that are consistently activated across domains are selected, as such activation patterns indicate cross-domain universal knowledge closely tied to model generalization. To transform the selected experts into a compact and reusable form, XPERT further introduces a knowledge consolidation step. As shown in Step 2 of Figure 2, experts from each transformer block are aggregated into high-order tensors, upon which tensor decomposition is applied to extract shared components that capture domain-agnostic expert knowledge.

A practical challenge arises when transferring the extracted knowledge to target models whose parameter dimensions differ from those of the source MoE LLMs. To address this mismatch, we propose a parameter-scale adaptation method that adjusts the extracted tensor representations to align

with the dimensionality of the target model. As illustrated in Step 3 of Figure 2, the reconstructed representations are adapted into Feed-Forward Network (FFN) parameter matrices that match the target model's hidden and intermediate dimensions, and are used to initialize its FFN layers prior to training. All remaining parameters are initialized using standard random initialization.

We evaluate XPERT using OLMoE-7B (Muennighoff et al., 2025) and DeepSeekMoE-16B (Dai et al., 2024) as source MoE LLMs on supervised fine-tuning (SFT) benchmarks spanning language understanding and dialogue generation across multiple domains. Across all settings, XPERT extracts a compact subset of expert-derived representations (approximately 1.25% of the parameters in source LLMs). Notably, expert selection, knowledge consolidation, and parameter-scale adaptation are entirely training-free. When used as an initialization prior, XPERT-initialized models consistently outperform strong baselines, including training from scratch and knowledge distillation, while converging faster and requiring up to 5× less pre-training data. Overall, these results demonstrate that expert knowledge in MoE LLMs can be systematically reused as an effective and generalizable initialization prior. XPERT highlights the potential of MoE LLMs as structured knowledge sources beyond their original deployment. Our contributions can be summarized as follows:

- We systematically study the reuse of expert knowledge in MoE LLMs, and show that a subset of experts consistently encodes cross-domain universal knowledge closely tied to model generalization.

- We propose XPERT, a framework that extracts, consolidates, and adapts expert knowledge from pre-trained MoE LLMs to support more effective model training, enabling parameter-level reuse in models of diverse dimensions and sizes.

- Through extensive experiments across multiple model scales and diverse downstream tasks, we demonstrate that XPERT-initialized models exhibit improved training dynamics and stronger downstream performance compared to other baselines.

## 2. Related Work

**Mixture-of-Experts Language Model**   MoE language models have been widely studied as an efficient architecture for scaling model capacity through conditional computation, where only a subset of experts is activated for each input (Shazeer et al., 2017b). Subsequent work has explored large-scale MoE pre-training and deployment, focusing on efficiency, scalability, and expert specialization, including Switch Transformer (Fedus et al., 2022), GShard (Lepikhin

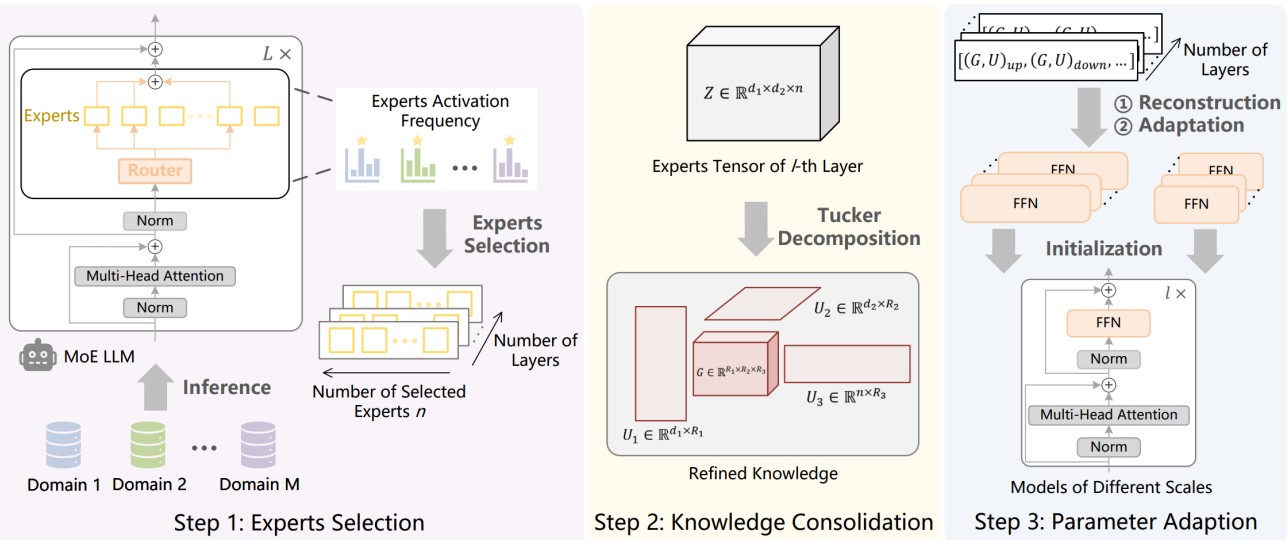

*Figure 2.* The framework of XPERT. $Z$ is the tensor formed by stacking the selected experts parameter matrices, and $(\mathcal{G}, U)$ represents the refined knowledge corresponding to a specific parameter matrix. After parameter-scale adaptation in Step 3, the extracted expert knowledge is used to initialize the FFN layers of language models with different scales.

et al., 2020), and more recent MoE-based LLMs such as DeepSeekMoE (Dai et al., 2024), OLMoE (Muennighoff et al., 2025), and Gemini (Team et al., 2024). Several studies further analyze expert routing behaviors and activation patterns to understand specialization and load balancing (Fedus et al., 2022; Huang et al., 2024). However, existing work primarily examines experts in the context of routing efficiency and task specialization, and does not investigate whether expert parameters encode reusable knowledge that can be systematically extracted and transferred to new models.

**Reusable Knowledge in Models**    Prior work has explored whether knowledge embedded in large models can be explicitly identified and reused (Liu et al., 2026; Shi et al., 2026a). Some studies estimate parameter or component importance through gradient sensitivity (Molchanov et al., 2019), loss-change tracking (Frankle & Carbin, 2018), or data valuation methods (Xu et al., 2024). Recent studies on LLM pruning further show that removing model components can induce substantial representation shifts and performance collapse, highlighting the importance of understanding which structures preserve model capability (Shi et al., 2026b). While effective at highlighting influential parameters, these approaches typically reveal patterns that are either highly dispersed or aggregated at coarse granularities, limiting their suitability as reusable knowledge units.

Other work focuses on parameter-efficient adaptation, such as adapters and low-rank updates (Houlsby et al., 2019; Hu et al., 2022; Li & Liang, 2021), which improve training efficiency but do not explicitly isolate transferable, task-agnostic knowledge representations. Overall, existing ap-

proaches offer limited insight into how generalizable knowledge can be explicitly localized and reused across models. Whether LLMs contain structured, reusable knowledge components remains an open question, particularly for MoE architectures, where knowledge is explicitly modularized at the expert level.

**Model Initialization**    Model initialization strongly influences optimization dynamics and generalization in deep neural networks. While classical schemes such as Xavier and Kaiming focus on training stability (Glorot & Bengio, 2010; He et al., 2015), modern language models often rely on pretrained initialization to reuse representations learned from large-scale data, leading to faster convergence and improved performance (Radford et al., 2019; Devlin et al., 2019). Prior studies further show that initialization biases which parameter substructures are activated during training and can steer optimization toward better-generalizing solutions (Frankle & Carbin, 2018; Mishkin & Matas, 2015). XPERT builds on this insight by reusing expert knowledge from MoE LLMs at initialization, providing a structured inductive bias that improves training effectiveness. Beyond conventional random initialization, recent work has explored knowledge-aware initialization strategies for model scaling, including bidirectional knowledge transfer and frequency-domain knowledge reuse (Shen et al., 2026).

## 3. Expert Knowledge Transfer Framework

We present **XPERT**, a framework that reuses expert knowledge from pre-trained MoE LLMs to initialize new models for improving training effectiveness. XPERT consists of

three stages: cross-domain expert selection (Section 3.2), expert knowledge consolidation via tensor decomposition (Section 3.3), and parameter-scale adaptation for initialization (Section 3.4).

## 3.1. Preliminary

The MoE architecture consists of multiple experts $E$ and a router $G(x)$. The input data are processed through the router, which selects a subset of experts for computation. In an MoE layer with $N$ experts, each token $x$ is assigned a subset of experts for computation through the router:

$$y = \sum_{i=1}^{N} G_i(x)E_i(x),$$

$$G_i(x) = \begin{cases} g_i, & \text{if } g_i \in TopK(\{g_j \mid 1 \leq j \leq N\}, K), \\ 0, & \text{otherwise}, \end{cases}$$

where $E_i$ represents the $i$-th expert, which is implemented as the FFN layer in a Transformer model. The term $g_i = \text{softmax}_i(Wx)$ denotes the softmax value obtained for the $i$-th expert after the gating mechanism processes the input $x$. $W$ represents the weight parameters of the gating function.

In some recent MoE LLMs (Dai et al., 2024; Liu et al., 2024), shared experts have been introduced to handle common knowledge in different tasks. The calculation of an MoE architecture with shared experts is formulated as $y = \sum_{i=1}^{N} G_i(x)E_i(x) + \sum_{i=1}^{S} E_i^s(x)$, where $E^s$ represents the shared experts, $S$ is the number of the shared experts.

## 3.2. Cross-domain Experts Selection

As discussed in the Introduction, we aim to identify experts that encode cross-domain knowledge closely tied to model generalization. Such experts are expected to be consistently activated across inputs from multiple domains, indicating that they capture knowledge shared beyond a single task or domain. Accordingly, the first step of XPERT is to identify common experts with strong cross-domain activation patterns. In certain MoE architectures, such as DeepSeekMoE, shared experts are explicitly introduced and activated across all inputs. We directly treat these shared experts as common experts. For more general MoE settings where shared experts are not predefined, we introduce an automatic selection criterion based on expert activation statistics.

As shown in Step 1 of Figure 2, we begin by utilizing $M$ different domain-specific datasets $D_1, D_2, \ldots, D_M$, each corresponding to a different domain. We define the activation probability of the expert $i$ on the dataset $D_m$ as $P_{i,m}$. $P_{i,m}$ reflects the usage probability of the $i$-th expert on domain $m$, that is, the proportion of the number of times the expert $i$ is activated to the total number of activation of all experts.

To identify experts that are active across multiple domains, we compute the average activation level of expert $i$ across all datasets:

$$A_i = \frac{1}{M} \sum_{m=1}^{M} P_{i,m}. \tag{1}$$

In the above function, a higher $A_i$ indicates that expert $i$ is frequently activated across multiple domains, suggesting that it encapsulates a greater amount of shared knowledge among domains. Furthermore, to avoid selecting experts that exhibit high activation in only a few domains while remaining inactive in others, we also consider the consistency score, which measures the balance of an expert's activation across different domains. A lower consistency score of expert $i$ implies that the activation of expert $i$ is relatively balanced across all the domains, rather than being highly responsive to only a few domains. The consistency score of the expert $i$ is:

$$C_i = \frac{1}{M} \sum_{m=1}^{M} |P_{i,m} - A_i|. \tag{2}$$

Finally, by combining the average activation level of experts across all datasets with their activation consistency, we identify the experts with cross-domain common knowledge: $\varepsilon = TopK(\{A_i - C_i | 1 \leq i \leq N\}, n)$, where $n$ represents the number of experts with common knowledge to be selected. $\varepsilon$ is the set of the selected experts with common knowledge.

## 3.3. Generalizable Expert Knowledge Consolidation

While the experts we selected are good at handling common knowledge across domains, the shared generalizable knowledge they contain still requires further refinement. To preserve higher-purity shared knowledge from these experts, we need to further keep the common information between them and remove their differences, which helps create cleaner generalization knowledge.

Specifically, we employ Tucker decomposition (Tucker, 1966) to extract shared structures from the parameters of the selected experts, aiming to refine generalizable knowledge. Tucker decomposition factorizes a high-dimensional tensor into a core tensor and several factor matrices, retaining only partial information along each dimension. The core tensor obtained through Tucker decomposition captures the relationships between the principal components across different dimensions of the original tensor. By retaining only a small number of principal components, the impact of noise from individual experts is reduced, allowing the core tensor to effectively capture key commonalities among experts (more detailed analysis is given in the Appendix D).

First, we stack the expert matrices together to form a higher-order tensor $Z \in \mathbb{R}^{d_i \times d_o \times n}$. Here, $d_i$ and $d_o$ are the dimensions of the expert matrices from the LLM. For example, $d_i$ could be the embedding dimension in the FFN layer, and $d_o$ might be the intermediate size in the same FFN layer.

As illustrated in Step 2 of Figure 2, we employ the Tucker decomposition to the tensor $Z$:

$$\mathcal{G},\ U = Tucker(Z,\ (R_1,\ R_2,\ R_3)), \tag{3}$$

where $\mathcal{G} \in \mathbb{R}^{R_1 \times R_2 \times R_3}$ represents the core tensor, $U = (U_1,\ U_2,\ U_3)$ and $U_i$ denotes the factor matrix along the $i$-th dimension. The values $R_1$, $R_2$, and $R_3$ are hyperparameters that determine the rank of the principal components retained in each dimension. The resulting core tensor and factor matrices together form a compact representation of generalization and common knowledge, refined from the selected experts. In practice, this consolidated representation accounts for only a small fraction of the parameters of the source MoE model (e.g., approximately **1.25%** in OLMoE-7B), while preserving the dominant generalizable structures shared across experts.

### 3.4. Parameter-scale Adaptation and Initialization

After refining common expert knowledge through tensor decomposition, the final step of XPERT is to adapt the consolidated representation to the parameter dimensions of target models and use it for initialization. This step ensures that expert knowledge extracted from a source MoE LLM can be flexibly reused across models with different dimensions and sizes.

As illustrated in Step 3 of Figure 2, the consolidation step produces a core tensor $\mathcal{G} \in \mathbb{R}^{R_1 \times R_2 \times R_3}$ and factor matrices $U = (U_1, U_2, U_3)$. Since the third mode corresponds to the expert dimension, we first aggregate expert-specific components by averaging: $\bar{U}_3 = 1/n \sum_{i=1}^{n} U_3[:, i]$, where $\bar{U}_3 \in \mathbb{R}^{R_3 \times 1}$. We then reconstruct the consolidated FFN parameter matrix via mode-wise multiplication:

$$\Gamma = \mathcal{G} \times_1 U_1 \times_2 U_2 \times_3 \bar{U}_3, \tag{4}$$

where $\Gamma \in \mathbb{R}^{d_i \times d_o}$. For each block $l$ and FFN projection $j$, the reconstructed matrix $\Gamma_{l,j}$ is then resized via parameter-scale adaptation to match the target FFN dimensions and then used for FFN initialization.

To operationalize this resizing step, we introduce a parameter-scale adaptation mechanism that maps a reconstructed matrix to arbitrary target FFN dimensions while preserving its dominant structure. A naive approach would directly truncate rows or columns when reducing dimensionality, or replicate and pad parameters when expanding dimensionality. However, such operations ignore the internal organization of the parameter matrix and can severely disrupt the structural patterns and semantic information encoded in the weights, leading to substantial degradation in downstream performance.

Instead, we design a structure-preserving adaptation mechanism that aims to retain as much informative content as possible while flexibly adjusting matrix size. Concretely, given a reconstructed matrix $\Gamma \in \mathbb{R}^{d_i \times d_o}$ and target dimensions $(d_i', d_o')$, we adapt it to $\hat{M} \in \mathbb{R}^{d_i' \times d_o'}$ in two stages: (i) low-rank factorization with importance scoring to identify structurally salient components, and (ii) importance-guided row and column resampling that contracts or expands the matrix while preserving its dominant subspace. This design enables XPERT to reuse expert-derived knowledge across models of different sizes without destroying the underlying parameter structure.

**Stage 1. Factorization and scoring** We perform a truncated singular value decomposition (SVD) on $\Gamma$: $\Gamma \approx U\Sigma V^\top$, where $U \in \mathbb{R}^{d_i \times r}$, $\Sigma \in \mathbb{R}^{r \times r}$, $V \in \mathbb{R}^{d_o \times r}$, and $r$ is the number of retained principal components. We compute importance scores for rows of $U$ and $V$:

$$s_i^{(U)} = |u_i|_2,\ i = 1, \ldots, d_i; \quad s_j^{(V)} = |v_j|_2,\ j = 1, \ldots, d_o,$$

where $u_i$ denotes the $i$-th row of $U$ and $v_j$ denotes the $j$-th row of $V$.

**Stage 2. Resampling to target size** For the row dimension, when $d_i' \leq d_i$, we select the indices of the top-$p$ rows by score:

$$\mathcal{I} = \text{TopK}(\{s_i^{(U)}\}, d_i'),\ \ U' = U[\mathcal{I}, :] \in \mathbb{R}^{d_i' \times r}.$$

When $d_i' > d_i$, we replicate the most important $(d_i' - d_i)$ rows and append them:

$$\mathcal{I} = \text{TopK}(\{s_i^{(U)}\}, d_i' - d_i),\ \ U' = \begin{bmatrix} U \\ U[\mathcal{I}, :] \end{bmatrix} \in \mathbb{R}^{d_i' \times r}.$$

The row dimension is handled analogously using $s_j^{(V)}$ to obtain $V' \in \mathbb{R}^{d_o' \times r}$. Finally, we reconstruct the resized matrix as $\hat{M} = U'\Sigma V'^\top \in \mathbb{R}^{d_i' \times d_o'}$.

The resulting matrix $\hat{M}$ is then used to initialize the corresponding FFN weight matrix of the target model, while all remaining parameters are initialized with standard random initialization. This adaptation is entirely training-free and enables XPERT to reuse consolidated expert knowledge across models with different FFN dimensions.

## 4. Experiments

Our experiments aim to evaluate whether expert knowledge extracted from MoE LLMs can be effectively reused to support more efficient and robust model training across tasks

*Table 1.* Results of models with different scales on language understanding benchmarks. We evaluate models of different dimensions and depths. All baselines are first pre-trained on 5B tokens, and fine-tuned for 3 epochs on these datasets.

| #Params | #Size | Baseline | Commonsense & Reading Comprehension | | | | Law | Medicine | Avg. |
|---|---|---|---|---|---|---|---|---|---|
| | | | BoolQ | Hellaswag | WinoGrande | PIQA | CaseHold | MedMCQA | |
| 270M | dim=1024 16 Layers | Scratch | 71.56 | 25.21 | 49.64 | 51.96 | 80.93 | 32.35 | 51.94 |
| | | Distillation | 72.66 | 26.28 | 49.64 | 52.39 | 82.54 | 34.07 | 52.93 |
| | | XPERT-OLMoE | 73.21 | **26.99** | **50.75** | **54.46** | 83.11 | 35.05 | **53.93** |
| | | XPERT-DeepSeek | **73.24** | 26.53 | 50.67 | 53.43 | **83.13** | **35.12** | 53.69 |
| 391M | dim=2048 8 Layers | Scratch | 70.83 | 26.95 | 49.09 | 50.00 | 78.50 | 32.09 | 51.24 |
| | | Distillation | 71.32 | 26.21 | 49.41 | 50.22 | 80.48 | 32.30 | 51.66 |
| | | XPERT-OLMoE | **72.63** | **27.27** | **50.59** | 51.80 | **81.50** | 34.16 | **52.99** |
| | | XPERT-DeepSeek | 71.56 | 27.15 | 50.36 | **52.77** | 81.10 | **34.50** | 52.91 |
| 480M | dim=2048 12 Layers | Scratch | 71.56 | 26.80 | 48.86 | 51.90 | 80.97 | 32.78 | 52.14 |
| | | Distillation | 71.62 | 26.40 | 49.17 | 50.87 | 81.35 | 32.39 | 51.97 |
| | | XPERT-OLMoE | **73.76** | **27.40** | **50.36** | **55.28** | **82.90** | 35.04 | **54.12** |
| | | XPERT-DeepSeek | 72.35 | 26.47 | 50.12 | 51.25 | 82.12 | **35.50** | 52.97 |
| 570M | dim=2048 16 Layers | Scratch | 70.86 | 26.29 | 48.62 | 51.89 | 80.16 | 33.04 | 51.81 |
| | | Distillation | 71.77 | 26.47 | 48.86 | 52.68 | 81.35 | 33.52 | 52.44 |
| | | XPERT-OLMoE | **73.91** | **27.56** | 49.96 | **55.44** | **82.96** | **34.78** | **54.10** |
| | | XPERT-DeepSeek | 72.14 | 27.44 | **50.75** | 54.84 | 82.54 | 34.47 | 53.70 |

and domains. We assess XPERT from two perspectives in datasets: **task generality**, covering both language understanding and dialogue generation, and **domain generality**, spanning multiple vertical knowledge domains. Across all experiments, we focus on comparing different initialization strategies under controlled training settings.

### 4.1. Baselines and Datasets

**Baselines.** XPERT-initialized models are dense language models following the Llama architecture, instantiated with varying model dimensions and depths. To ensure fair comparison, all baselines adopt the same model architecture and training configuration. We compare XPERT with representative baselines that differ in how external knowledge is introduced into the model under an identical architecture and training pipeline. **Scratch** trains the target model from standard random initialization without any external knowledge transfer. **Distillation** (Gu et al., 2024) pre-trains the target model from random initialization using knowledge distillation from a source LLM (OLMoE-7B). This baseline serves as a control to examine whether the benefits of XPERT can be attributed solely to transferring knowledge through teacher–student supervision, rather than through parameter-level knowledge reuse. **XPERT-OLMoE** initializes the target model using expert knowledge extracted from OLMoE-7B (Muennighoff et al., 2025). **XPERT-DeepSeek** initializes the target model using expert knowledge extracted from DeepSeekMoE-16B (Dai et al., 2024).

For fair comparison, models with the same scale have identical parameter counts and are pre-trained on the same number of tokens. After pre-training, each model is fine-tuned independently on individual SFT datasets, allowing evaluation

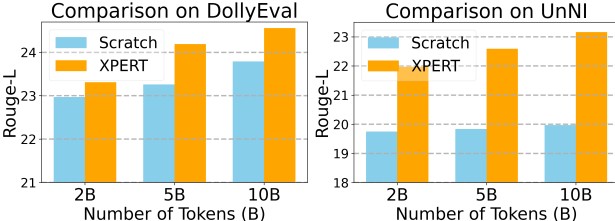

*Figure 3.* Comparison of fine-tuning performance between 16-layer XPERT-OLMoE and Scratch under varying pre-training data budgets (2B, 5B, and 10B tokens).

on task-specific performance and training dynamics.

**Datasets.** We evaluate XPERT on a diverse set of SFT tasks defined along two dimensions: task type and knowledge domain. For language understanding, we adopt widely used benchmarks including **BoolQ** (Clark et al., 2019), **HellaSwag** (Zellers et al., 2019), **PIQA** (Bisk et al., 2020), **WinoGrande** (Sakaguchi et al., 2021), **CaseHold** (Zheng et al., 2021), and **MedMCQA** (Pal et al., 2022), which cover reasoning, commonsense, and domain-specific understanding. For dialogue generation, we follow the evaluation setup of MiniLLM (Gu et al., 2024) and evaluate on **Dolly**[1], **S-NI** (Wang et al., 2022), **UnNI** (Honovich et al., 2023), **SelfInst** (Wang et al., 2023b), and **Vicuna** (Chiang et al., 2023).

### 4.2. Main Results

To comprehensively evaluate the effectiveness of XPERT in reusing expert knowledge for model training, we con-

---

[1]https://github.com/databrickslabs/dolly/tree/master

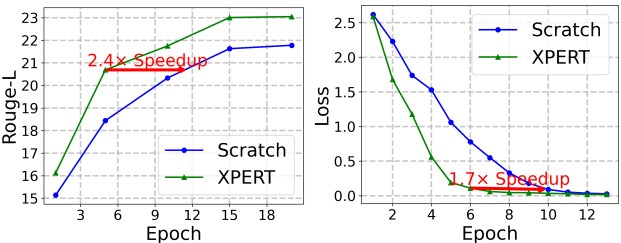

*Figure 4.* Comparison of Rouge-L and loss curves between Scratch and XPERT-initialized models on the DollyEval dataset.

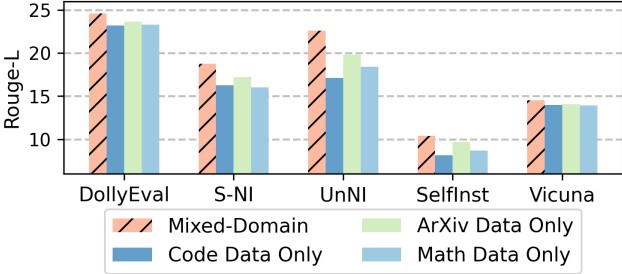

*Figure 5.* Effect of domain diversity in expert selection on downstream SFT performance of XPERT-initialized models. **Mixed-Domain** uses a diverse corpus spanning multiple knowledge domains, including (but not limited to) Wikipedia, GitHub, and arXiv.

duct experiments from three perspectives: **downstream task performance**, **fine-tuning convergence behavior**, and **robustness under different pre-training budgets**.

**Superior performance across downstream benchmarks.** Table 1 and Table 2 summarize the results on language understanding and dialogue generation benchmarks, respectively. Across a broad range of tasks and domains, models trained with XPERT consistently outperform strong baselines, including Scratch and Distillation. For instance, on the S-NI generation task, the 16-layer XPERT model (570M) improves over Scratch by **4.21** and over Distillation by **3.76**.

Prior work shows that knowledge distillation degrades when there is a large capacity gap between teacher and student models (Mirzadeh et al., 2020; Gao et al., 2020; Wang & Yoon, 2021). In our experiments, we use OLMoE-7B as the teacher model for all distillation baselines to eliminate confounding factors from different source models with XPERT baselines. Under the same teacher and training setup, XPERT achieves consistently better results than knowledge distillation.

This suggests that the knowledge encoded in MoE LLMs is difficult to transfer effectively through supervision alone. In contrast, XPERT reuses knowledge at the expert parameter level by explicitly identifying and consolidating expert representations closely related to generalization. This enables more precise and stable knowledge transfer that better matches the capacity of the target model. In addition, XPERT avoids repeated forward passes through the teacher model, resulting in substantially lower training cost compared to distillation. Additional results are provided in Appendix C.

**Faster convergence during fine-tuning.** To examine how expert knowledge reuse influences optimization behavior, we compare the fine-tuning dynamics of XPERT-OLMoE and Scratch. Figure 4 presents the Rouge-L scores and training loss curves on the Dolly dataset for 16-layer models (570M). Models trained with XPERT reach the same Rouge-L performance approximately **2.4×** earlier than Scratch and achieve comparable loss values about **1.7×** faster. This accelerated convergence indicates that reusing expert-level knowledge shapes a more favorable optimization landscape, allowing the model to exploit training signals more efficiently from the early stages of fine-tuning.

The faster convergence observed during downstream fine-tuning indicates that the reused cross-domain expert knowledge provides a strong and well-aligned inductive bias, enabling models to adapt more efficiently to task-specific supervision. By initializing models with expert representations that already encode generalizable structure, XPERT reduces the amount of optimization needed to rediscover such knowledge during SFT, leading to faster convergence and improved training efficiency.

**XPERT reduces the amount of pre-training data needed to reach the same performance.** Figure 3 reports fine-tuning results on the UnNI dataset for 16-layer 570M models pre-trained with different amounts of data, while keeping the model scale and training pipeline fixed. Across all settings, XPERT-OLMoE consistently outperforms the Scratch baseline. Notably, on the UnNI dataset, a Scratch model pre-trained with 10B tokens fails to match the performance of XPERT-OLMoE pre-trained with only 2B tokens. These results indicate that XPERT substantially lowers the amount of pre-training data required to reach a given downstream performance under identical training pipelines. Compared with Scratch, XPERT provides a stronger starting point that enables the model to benefit more effectively from the same training signal, leading to performance gains comparable to several-fold increases in pre-training data. Additional results across tasks are provided in Appendix E.

### 4.3. Ablation Studies

**Component-wise Ablation of XPERT** Table 3 presents ablation results for the core components of XPERT:

- **Expert selection.** *Top-1 expert* leads to clear performance degradation, indicating that reusing a single expert is insufficient to capture the shared structure underlying cross-domain generalizable knowledge. *Ran-*

*Table 2.* Results of models with different scales on dialogue generation benchmarks. For evaluating generative capabilities of models, models (pre-trained on 5B tokens) are first fine-tuned on the Dolly dataset, and evaluated on the datasets shown in the table.

| #Params | #Size | Baseline | Dialogue Generation | | | | | Avg. (Rouge-L) |
| --- | --- | --- | --- | --- | --- | --- | --- | --- |
| | | | DollyEval | S-NI | UnNI | SelfInst | Vicuna | |
| 270M | dim=1024 16 Layers | Scratch | 21.31 | 14.67 | 19.32 | 9.49 | 13.77 | 15.73 |
| | | Distillation | 22.43 | 15.00 | 20.50 | 9.26 | 13.52 | 16.14 |
| | | XPERT-OLMoE | **23.67** | **16.58** | **21.55** | 8.44 | **14.46** | **16.94** |
| | | XPERT-DeepSeek | 23.22 | 15.20 | 21.33 | **9.56** | 13.25 | 16.51 |
| 391M | dim=2048 8 Layers | Scratch | 22.94 | 15.69 | 18.16 | 8.50 | 14.25 | 15.91 |
| | | Distillation | 22.78 | 16.63 | 18.14 | 8.34 | 13.01 | 15.78 |
| | | XPERT-OLMoE | 23.17 | **17.39** | **21.47** | **9.18** | **15.05** | **17.25** |
| | | XPERT-DeepSeek | **23.35** | 17.34 | 20.02 | 8.72 | 13.99 | 16.68 |
| 480M | dim=2048 12 Layers | Scratch | 22.48 | 16.57 | 20.76 | 9.41 | 14.20 | 16.68 |
| | | Distillation | 22.90 | 15.70 | 21.47 | 9.01 | 14.39 | 16.69 |
| | | XPERT-OLMoE | 23.36 | **19.43** | **21.83** | 9.57 | **14.83** | **17.80** |
| | | XPERT-DeepSeek | **23.38** | 18.01 | 20.48 | **10.30** | 13.92 | 17.22 |
| 570M | dim=2048 16 Layers | Scratch | 22.86 | 15.61 | 19.67 | 8.31 | 13.42 | 15.97 |
| | | Distillation | 22.09 | 16.06 | 20.27 | 10.20 | 14.09 | 16.54 |
| | | XPERT-OLMoE | **24.19** | **19.82** | 22.60 | **11.31** | **14.57** | **18.50** |
| | | XPERT-DeepSeek | 23.42 | 17.99 | **23.20** | 10.43 | 14.37 | 17.88 |

*Table 3.* Ablation results for different modules in XPERT. For each ablation variant, only the specified component is modified, while all other components follow the same configuration as the full XPERT pipeline. WinoG. denotes the Winogrande dataset.

| Method | BoolQ | Hellaswag | PIQA | WinoG. | Avg. |
| --- | --- | --- | --- | --- | --- |
| **Experts Selection Ablation** | | | | | |
| *Top-1 expert* | 71.68 | 26.71 | 52.07 | 48.86 | 49.83 |
| *Random Selection* | 71.89 | 26.97 | 53.32 | 48.46 | 50.16 |
| **Knowledge Consolidation Ablation** | | | | | |
| *Average Aggregation* | 72.02 | 26.73 | 51.41 | 48.22 | 49.60 |
| *CP Decomposition* | 70.70 | 25.81 | 51.58 | 49.72 | 49.45 |
| *SVD* | 70.09 | 26.72 | 52.34 | 49.57 | 49.68 |
| **Parameter-scale Adaptation Ablation** | | | | | |
| *Truncating* | 71.35 | 24.73 | 52.12 | 49.33 | 49.38 |
| **XPERT** | **73.91** | **27.56** | **55.44** | **50.75** | **51.92** |

*dom Selection* also results in degraded performance, as it introduces domain-specialized experts whose representations encode task-specific patterns instead of cross-domain universal structure.

- **Knowledge consolidation.** Replacing Tucker decomposition with simpler or alternative strategies, including *Average Aggregation*, *CP Decomposition*, or *SVD*, consistently reduces performance. Direct averaging retains expert-specific variations and noise, while CP and SVD impose more restrictive factorization forms that fail to preserve the multi-dimensional shared structure across experts and parameter dimensions, resulting in weaker initialization.

- **Parameter-scale adaptation.** Removing the proposed adaptation mechanism and directly truncating reconstructed parameters to match target dimensions

(***Truncating***) also harms performance, confirming that naive resizing disrupts the structural integrity of expert representations.

Overall, XPERT achieves the best results across all tasks, demonstrating that cross-domain expert selection, knowledge consolidation, and parameter-scale adaptation jointly play a critical role in effective expert knowledge reuse.

**Effect of Domain Choice for Expert Selection** We study how the choice of data domains influences expert selection by identifying experts using different types of domain-specific inputs (Figure 5). The results indicate that **Mixed-Domain**, which is the expert selection strategy adopted by XPERT, consistently leads to stronger downstream SFT performance than using data from any single domain. Selecting experts based on heterogeneous data encourages the identification of experts that are active across domains, which are more likely to encode shared, domain-agnostic knowledge relevant to generalization. In contrast, experts selected from a single domain tend to over-specialize, resulting in weaker transfer across downstream tasks. These findings underscore the importance of domain diversity in expert selection for extracting reusable expert knowledge.

**Impact of the Number of Experts and Decomposition Rank** Table 4 shows that the number of selected experts has a noticeable impact on downstream performance. Selecting a moderate number of experts yields the best results: using $K = 8$ experts consistently outperforms smaller settings, while further increasing the number of experts to $K = 16$ leads to degraded performance. This non-monotonic trend suggests that selecting too many experts may introduce domain-specific or task-dependent information, which di-

*Table 4.* Downstream performance under different numbers of selected common experts and Tucker ranks.

| Top-K Selected | #Rank | BoolQ | Hellaswag | PIQA | WinoG. | Avg. |
|---|---|---|---|---|---|---|
| 2 | (256, 512, 2) | 73.31 | 27.34 | 52.07 | 48.70 | 50.36 |
|   | (512, 1024, 2) | 72.05 | 27.40 | 52.94 | 48.86 | 50.31 |
| 8 | (256, 512, 8) | 73.91 | 27.56 | 55.44 | 49.96 | 51.71 |
|   | (512, 1024, 8) | 73.00 | 27.66 | 53.90 | 52.33 | 51.72 |
| 16 | (256, 512, 16) | 71.90 | 28.23 | 53.97 | 49.64 | 50.93 |
|   | (512, 1024, 16) | 73.06 | 28.07 | 52.47 | 50.04 | 50.91 |

lutes the purity of the extracted, generalizable knowledge.

The effect of the Tucker rank is comparatively less pronounced. Moderate rank settings are sufficient to preserve task-relevant representations, while overly large ranks provide limited additional benefit. Overall, these results suggest that effective expert reuse benefits from selecting a moderate number of experts and using compact factorization.

## 5. Conclusion

In this work, we show that MoE LLMs contain experts that are consistently activated across domains, revealing cross-domain, generalizable-relevant expert knowledge. Building on this observation, we propose XPERT, a training-free framework that extracts and reuses such expert knowledge to support effective model training across different scales. Experiments demonstrate that expert knowledge reuse improves downstream performance and accelerates convergence, highlighting the potential of MoE LLMs as structured and reusable knowledge sources.

## Acknowledgments

This research was supported by the Jiangsu Science Foundation (BK20243012, BG2024036), the National Science Foundation of China (62125602, U24A20324, 92464301), the New Cornerstone Science Foundation through the XPLORER PRIZE, and the Fundamental Research Funds for the Central Universities (2242025K30024). This work is also supported by National Natural Science Foundation of China (62576091); the Southeast University Big Data Computing Center; Southeast University Kunpeng & Ascend Center of Cultivation.

## Impact Statement

This paper presents work whose goal is to advance the understanding of expert knowledge reuse in Mixture-of-Experts language models and to improve training efficiency of downstream models with diverse scales. As with other advances in language model training, this work may have broader societal implications related to the use of language technologies. However, we do not identify any ethical concerns or societal impacts that are unique to the methods proposed in this paper beyond those already well established in prior research on large language models.

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

## A. Experts Frequency in OLMoE-7B

In the Introduction 1, we present the activation frequency of experts in the FFN layer of Layer 15 in the OLMoE-7B model across different domains (Github, Tulu, and Arxiv). Here, we provide additional visualizations. Figure 6 shows the activation frequencies of experts in Layer 0 and Layer 7 of the OLMoE-7B model across various domains. It can be observed that different experts within the same layer exhibit similar activation trends across these domains. For example, Expert 0, 21, and 40 in Layer 0, as well as Expert 2, 4, 17, 35, and 58 in Layer 7, all maintain high activation across multiple domains. This phenomenon indicates that MoE architectures inherently suffer from load imbalance among experts. Experts that are consistently activated across domains likely encode knowledge shared by multiple domains. We consider these experts to contain common knowledge that can be applied to a variety of tasks.

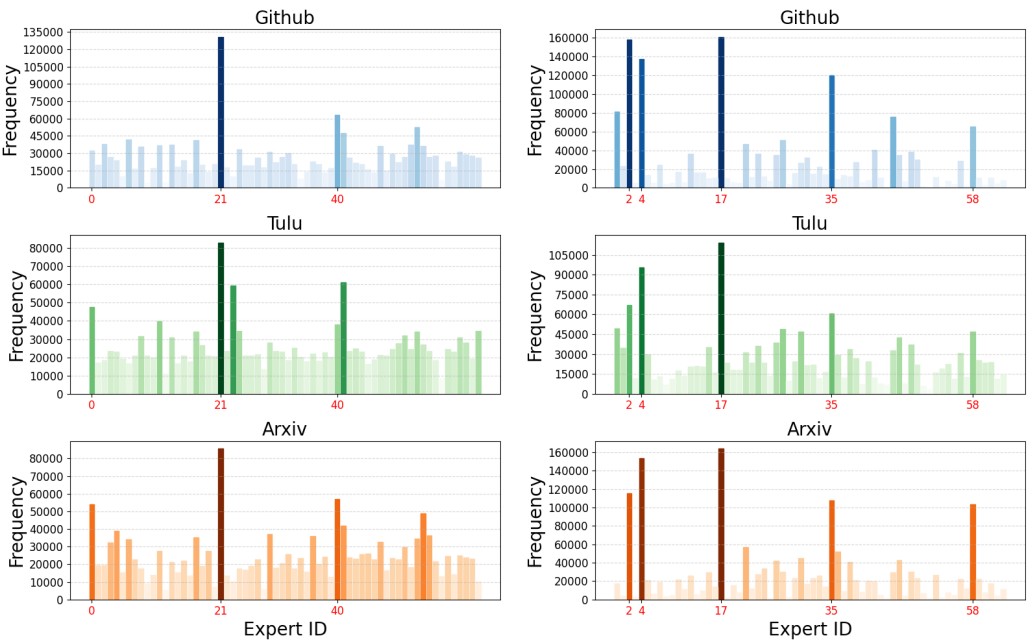

*Figure 6.* Expert activation frequencies of layer 0 (left) and layer 7 (right) in OLMoE-7B across different domains.

## B. Experimental Details

We set the embedding dimension of the target models to 2048, with an FFN intermediate size of 1024. To evaluate the effect of model scale, we experiment with models of 8, 12, and 16 transformer layers. Since expert knowledge is extracted independently from each block of the source MoE LLM, the resulting representations are organized in a block-wise manner. Accordingly, models with different depths are initialized using expert-derived parameters extracted from the corresponding number of blocks in the source model (e.g., an 8-layer model uses parameters from the first 8 blocks). All baseline models are subsequently trained using identical pre-training procedures and the same number of tokens to ensure fair comparison. After pre-training, the models are further fine-tuned on the aforementioned SFT datasets for evaluation.

For fine-tuning on the BoolQ, Hellaswag, MMLU, PIQA, and WinoGrande language understanding datasets, all baselines are trained for only 3 epochs. For evaluating generative capabilities of models, we follow the MiniLLM setting: models are first trained on the Dolly dataset for 100 epochs, and then evaluated on the DollyEval, S-NI, UnNI, SelfInst, and VicunaEval datasets.

**Datasets and code.** We use the RedPajama-V2 as our pre-training data, and the fine-tuning datasets for downstream tasks are presented in Section 4.

**Software Environment.** All experiments were conducted using PyTorch 2.1 with CUDA version 12.1 and 4×H100 GPUs.

**Model Configuration.** The LLMs are OLMoE-7B and DeepSeekMoE-16B, and the target models are Llama models (dense) with different sizes.

**Baseline Design Details.** In addition to the baselines based on XPERT (**XPERT-OLMoE** and **XPERT-DeepSeek**), we also include **Scratch** and **Distillation** as comparative baselines. Specifically, **Scratch** refers to a model that is randomly initialized, pre-trained, and then fine-tuned on downstream tasks. **Distillation** denotes a model that undergoes pre-training with distillation from the LLM, followed by fine-tuning on downstream tasks. To ensure the fairness of our comparisons, all baselines are configured with identical model parameter sizes and are pre-trained on datasets of the same scale.

**Training Hyperparameters.** The models are pre-trained using AdamW with an initial learning rate of 4e-4, batch size of 64, and a total of 20k-40k pre-training steps. During fine-tuning, we set the learning rate to 3e-4, and batch size to 8.

## C. More Results with Different Amount of Pre-training Data

In this section, we present the fine-tuning results of various baselines on downstream datasets after pre-training with different amounts of data in Table 5 and Table 6. It can be observed that XPERT consistently outperforms other baselines across different tasks, demonstrating a significant advantage.

*Table 5.* Results of models with different scales on model generation benchmarks (2B tokens pre-trained).

| Model | Baseline | DollyEval | S-NI | UnNI | SelfInst | VicunaEval |
|---|---|---|---|---|---|---|
| 8 Layers 391M | Scratch | 21.95 | 15.46 | 19.33 | 8.71 | 13.51 |
| | Distillation | 21.67 | 15.47 | 17.94 | 7.58 | 13.24 |
| | XPERT-OLMoE | 22.64 | **16.28** | 18.93 | **10.09** | 13.83 |
| | XPERT-DeepSeek | **22.66** | **16.28** | **20.17** | 9.35 | **14.60** |
| 12 Layers 480M | Scratch | 22.39 | 14.39 | 19.28 | 8.65 | 13.81 |
| | Distillation | 21.31 | 14.33 | **21.47** | 8.88 | 13.91 |
| | XPERT-OLMoE | **23.36** | **16.44** | 19.79 | **9.44** | 14.01 |
| | XPERT-DeepSeek | 22.76 | 14.32 | 19.06 | 9.38 | **14.16** |
| 16 Layers 570M | Scratch | 22.97 | 16.74 | 19.75 | 9.17 | 13.35 |
| | Distillation | 22.91 | 15.84 | 18.85 | 8.77 | 13.98 |
| | XPERT-OLMoE | **23.84** | **17.50** | **21.97** | 9.71 | **14.30** |
| | XPERT-DeepSeek | 23.31 | 16.71 | 20.89 | **10.26** | 13.24 |

*Table 6.* Results of models with different scales on model generation benchmarks (4B tokens pre-trained).

| Model | Baseline | DollyEval | S-NI | UnNI | SelfInst | VicunaEval |
|---|---|---|---|---|---|---|
| 8 Layers 391M | Scratch | 22.08 | 16.52 | 19.82 | 8.73 | 13.13 |
| | Distillation | 21.90 | 15.47 | 18.67 | 8.33 | 13.14 |
| | XPERT-OLMoE | 22.66 | **17.39** | **20.18** | **9.59** | **14.33** |
| | XPERT-DeepSeek | **22.70** | 17.04 | 19.74 | 8.29 | 13.77 |
| 12 Layers 480M | Scratch | 22.84 | 16.25 | 19.57 | 8.96 | 14.87 |
| | Distillation | 22.73 | 16.62 | 19.48 | 9.20 | 14.20 |
| | XPERT-OLMoE | 23.49 | **17.95** | **22.08** | 9.48 | **15.53** |
| | XPERT-DeepSeek | **23.76** | 17.40 | 20.24 | **10.43** | 14.79 |
| 16 Layers 570M | Scratch | 22.78 | 15.68 | 19.65 | 9.53 | 14.05 |
| | Distillation | 22.09 | 16.06 | 20.27 | 10.20 | 14.09 |
| | XPERT-OLMoE | 23.52 | **18.79** | **23.16** | 10.38 | **14.54** |
| | XPERT-DeepSeek | **23.90** | 17.30 | 21.93 | 11.05 | 14.19 |

## D. Analysis of Tucker Decomposition

In this section, we analyze the theoretical rationality behind using Tucker decomposition to extract shared knowledge among experts for refining the generalizable expert knowledge. To identify shared knowledge across a group of expert parameter

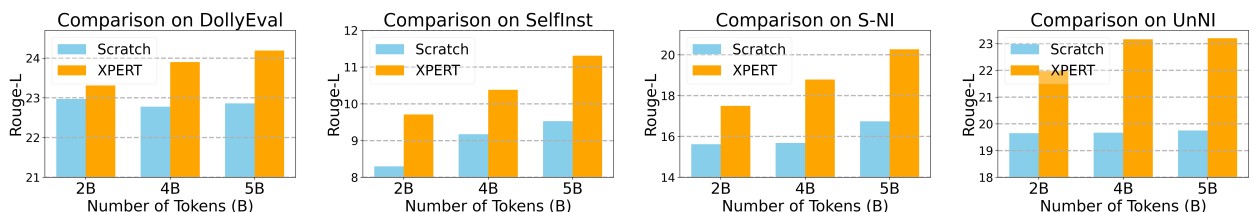

*Figure 7.* Comparison of fine-tuning performance between 16-layer GeneLLM-OLMoE and Scratch under varying pre-training token budgets

matrices, we first stack them into a third-order tensor and apply Tucker decomposition to extract compact representations along each dimension. Formally, given a set of matrices $\{X_i \in \mathbb{R}^{d_1 \times d_2}\}_{i=1}^n$, we construct a tensor $\mathcal{X} \in \mathbb{R}^{d_1 \times d_2 \times n}$ where $\mathcal{X}(:,:,i) = X_i$. This tensor captures both the internal structure of each matrix and the relationships across matrices.

We then perform Tucker decomposition:

$$\mathcal{X} \approx \mathcal{G} \times_1 U_1 \times_2 U_2 \times_3 \bar{U}_3, \tag{5}$$

where $\mathcal{G}$ is a compact core tensor and $U_1, U_2, U_3$ are orthogonal factor matrices capturing the most important components in each mode. By retaining only the top principal components (i.e., low-rank approximation), Tucker decomposition filters out high-frequency noise or task-specific variance while preserving the dominant, shared structures that are consistently present across the matrices.

Intuitively, if each matrix $X_i$ can be viewed as a combination of shared content $C$ and an instance-specific perturbation $\theta_i$, i.e., $X_i = C + \theta_i$, then stacking them as:

$$\mathcal{X} = \mathcal{C} + \Theta, \quad \mathcal{C}(:,:,i) = C, \ \Theta(:,:,i) = \theta_i. \tag{6}$$

Components with lower energy typically correspond to minor variations or idiosyncrasies specific to individual matrices, including random noise. By discarding these less significant components, the decomposition effectively denoises the representation and emphasizes structural coherence. If the matrices exhibit common patterns, such as similar structural characteristics or activation behaviors, these shared structures tend to manifest as high-energy components in the tensor. Tucker decomposition captures these components through the leading principal directions, allowing us to reconstruct the tensor while retaining information that is commonly embedded across matrices.

Since the decomposition optimizes the overall reconstruction error across expert matrices, the resulting low-rank representation preferentially preserves structural components that are consistently shared among experts, while suppressing task-specific variations and noise. As a result, the extracted representation emphasizes domain-general patterns that recur across experts rather than idiosyncratic behaviors tied to individual tasks or domains. This consolidated expert knowledge can then be reused as an effective initialization prior for target models, guiding subsequent training toward solutions with stronger generalization and improved data efficiency.

## E. Additional Results on the Effect of Pre-training Data Scale

In Section 4.2, we compare XPERT-initialized models with Scratch under different pre-training data scales and show that XPERT provides a strong advantage during subsequent training. In this section, we present additional results to further examine how this advantage manifests across a broader range of downstream tasks. Specifically, we evaluate the downstream performance of XPERT-OLMoE and Scratch after pre-training on varying amounts of data, ranging from 2B to 5B tokens, on multiple benchmarks including DollyEval, SelfInst, S-NI, and UnNI.

As shown in Figure 7, XPERT-initialized models consistently achieve stronger downstream performance than Scratch across all tasks and pre-training scales. Notably, models initialized with XPERT and pre-trained on fewer tokens can reach or exceed the performance of Scratch models trained with substantially more data under the same training procedure. These results indicate that the performance gains introduced by XPERT initialization are comparable to those obtained by increasing the scale of pre-training data, highlighting the effectiveness of expert-based initialization in improving how training signals are utilized.

## F. Robustness to Layer Selection in Expert-based Initialization

We further investigate whether the specific layer positions of the extracted expert knowledge play a critical role when initializing descendant models. If the expert-derived representations capture broadly generalizable knowledge, their effectiveness should not be overly sensitive to the exact layers from which they are inherited, as long as the structural order of layers is preserved. To examine this, we initialize an 8-layer descendant model using different layer selection strategies from the extracted expert knowledge: (1) randomly selecting 8 layers, (2) inheriting the first 8 layers, and (3) inheriting the last 8 layers. All other training settings are kept identical.

As shown in Table 7, the performance differences among different layer selection strategies are relatively small. Selecting layers in order generally leads to slightly better results than random selection, likely because it preserves the structural continuity between consecutive layers. Among the ordered strategies, using the first 8 layers yields marginally higher performance than using the last 8 layers, although the gap is not pronounced across downstream tasks. Overall, these results suggest that the expert knowledge extracted by XPERT is not strongly tied to specific layer indices, and that different reasonable layer selection strategies lead to comparable performance. In our implementation of XPERT, we therefore adopt a simple and consistent strategy by selecting expert knowledge from the shallow layers of the source MoE model in order, according to the depth of the target model.

*Table 7.* Effect of different layer selection strategies in XPERT when initializing an 8-layer descendant model. Results are averaged over Scratch, XPERT-OLMoE, and XPERT-DeepSeekMoE.

| Layer Selection Strategy (8 Layers) | BoolQ (Avg) | PIQA (Avg) | AVG |
|---|---|---|---|
| Random selection | 71.36 | 51.64 | 61.50 |
| First 8 layers | 72.10 | 52.29 | 62.20 |
| Last 8 layers | 72.05 | 51.66 | 61.86 |

*Table 8.* Sensitivity to learning rate for a 16-layer model on representative downstream tasks. Results are reported on BoolQ and PIQA.

| Learning Rate | Method | BoolQ | PIQA |
|---|---|---|---|
| | Scratch | 70.83 | 50.46 |
| 1e-5 | XPERT-OLMoE | 72.45 | 51.90 |
| | XPERT-DeepSeek | 72.05 | 51.52 |
| | Scratch | 70.86 | 51.89 |
| 3e-5 | XPERT-OLMoE | 73.91 | 55.44 |
| | XPERT-DeepSeek | 72.14 | 54.84 |
| | Scratch | 69.97 | 47.99 |
| 5e-5 | XPERT-OLMoE | 72.08 | 50.98 |
| | XPERT-DeepSeek | 72.05 | 50.33 |

## G. Hyperparameter Sensitivity Analysis

To ensure a fair comparison and rule out the possibility that performance gains stem from favorable hyperparameter choices, we conduct a grid search over key optimization hyperparameters for both XPERT and all baselines. Specifically, we vary the learning rate in {1e-5, 3e-5, 5e-5}, while keeping all other training settings identical. For each method, the best-performing configuration is selected based on development set performance.

Table 8 reports the results of a 16-layer model on representative downstream tasks. Across all learning rates, models initialized with XPERT consistently outperform the Scratch baseline and remain competitive or superior under different optimization settings. Notably, while baseline performance varies substantially with the learning rate, XPERT-initialized models exhibit more stable behavior and achieve strong results across a wider range of learning rates. These results indicate that the effectiveness of XPERT does not rely on carefully tuned hyperparameters, but rather arises from the quality of the expert-based initialization itself.

