# OpenReview forum: "XPERT: Expert Knowledge Transfer for Effective Training of Language Models"
_ICML.cc/2026/Conference — ICML 2026 regular_

### Official Review · Reviewer_MWRy · 2026-02-20

**Soundness:** 3
**Presentation:** 3
**Significance:** 3
**Originality:** 3
**Overall Recommendation:** 5
**Confidence:** 3

**Summary:**

This paper introduces XPERT, a training-free framework designed to extract and reuse generalizable knowledge from pre-trained Mixture-of-Experts LLMs to initialize dense language models. The authors identify "common experts" that are consistently activated across diverse domains and argue that these experts encode cross-domain universal knowledge. The framework employs Tucker decomposition for parameter extraction and a mechanism for adaptation. Empirical results across language understanding and dialogue tasks demonstrate that XPERT-initialized models achieve better performance, faster convergence, and higher data efficiency compared to training from scratch or traditional distillation baselines.

**Compliance With Llm Reviewing Policy:**

Affirmed.

**Final Justification:**

I thank the authors for their detailed rebuttal and for providing the additional experimental results.

1. The new experimental results are highly appreciated. They effectively address my concerns regarding the positioning of XPERT against other distillation and pruning methods. I look forward to seeing the more principled discussion on Tucker rank selection in the revised manuscript.

2. I acknowledge the authors' rationale for focusing on FFN layers. Given that current MoE architectures explicitly modularize knowledge within FFN blocks, targeting these for expert knowledge extraction is a sound and practical choice.

3. I have also read the response to Reviewer #wews regarding the distinction between "knowledge extraction" and "model compression." While I still consider the boundary between these two concepts to be somewhat blurred, I believe this conceptual nuance does not detract from the overall technical contribution and empirical value of the paper.

Overall, the authors have successfully addressed my questions. I am increasing my score to 5 (Accept).

**Key Questions For Authors:**

Could the authors provide more empirical comparisons with other baselines, specifically pruning or other distillation methods, to better position XPERT's performance?

Have the authors explored any methods to transfer knowledge for the non-MoE components (e.g., Attention layers) to adapt with the XPERT-initialized FFNs?

For "After pre-training, each model is fine-tuned independently on individual SFT datasets, allowing evaluation
on task-specific performance and training dynamics.", does this finetuning use the same hyper-parameter across different baselines?

**Limitations:**

Yes.

**Strengths And Weaknesses:**

Strengths:
1. The paper offers a novel perspective on MoE models as category reusable "knowledge sources" rather than just efficient inference architectures. The idea of using MoE expert activation patterns to initialize dense models is innovative.
2. The methodology is technically grounded, combining tensor decomposition with structural importance scoring. The experiments cover multiple source models and diverse downstream benchmarks.
3. XPERT is a training-free method to distill information from teacher models. This can reduce pre-training data requirements, which is highly relevant for the community, especially regarding efficient LLM pre-training. Open-source code is provided.


Weaknesses:
1. The current framework primarily focuses on initializing FFN layers. While FFNs contain a large portion of model parameters, the lack of a strategy for Attention layers means the model still relies on random initialization for a significant part of its architecture.
2. Although the paper compares against "Distillation", there are still distillation methods beyond MiniLLM. Espcially, as this method shares conceptual similarities with model pruning, a comparison against pruning baselines would be better.
3. While Table 4 explores Tucker ranks, the choice of ranks seems empirical. A more principled discussion on how to determine or search for optimal compression ratios for different model scales would strengthen the work.

---

> ### Author Rebuttal · Authors · 2026-03-30
>
> ****Response to Reviewer #MWRy****
> Thank you for your careful reading and insightful questions.
>
> ***Q1***   **Question1 & Weakness2**
>
> ***R1***   We agree that comparing against additional distillation and pruning baselines is important for better positioning our method.
>
> **(1) We have added new baselines under a controlled setting.**
>
> We include comparisons with **DistiLLM** and **EPP** (an expert pruning method). For fair comparison, all methods use the same **7B OLMoE source model** and ​**570M target model**​, follow the original settings of each method, and are evaluated after the same SFT procedure.
>
> | Baseline/Dataset | BoolQ | Winogrande | PIQA | Casehold|
> | --- | --- | --- |--- | --- |
> | XPERT | **73.91** | **50.75** | **55.44** | **82.96** |
> | DistiLLM [1] | 71.02 | 48.93 | 52.65 | 79.61 |
> | Pruning-EPP [2] | 63.82 | 48.38 | 50.54 | 62.37 |
>
> | Baseline/Dataset | DollyEval | UnNI |
> | --- | --- | --- |
> | XPERT | **24.19** | **22.60** |
> | DistiLLM [1] | 20.81 | 21.27 |
> | Pruning-EPP [2] | 18.59 | 11.41 |
>
> [1] DistiLLM: Towards Streamlined Distillation for Large Language Models, ICML2024
>
> [2] Efficient Expert Pruning for Sparse Mixture-of-Experts Language Models: Enhancing Performance and Reducing Inference Costs
>
> **(2) XPERT is fundamentally different from compression-based methods.**
>
> Both distillation and pruning are compression-oriented methods that aim to approximate or reduce the source model. In contrast, XPERT is **not** designed to compress the source model, but to **extract and reuse cross-domain expert knowledge** to improve the training of a new target model. We believe this distinction is important: rather than approximating the source model under a smaller budget, XPERT transfers structured expert knowledge, which leads to more effective learning under large source–target size mismatch.
>
> **ps.: We sincerely refer the reviewer to our response (R1) to Reviewer #wews for more details.**
>
> ***Q2***   **Question2 & Weakness1**
>
> ***R2***   We thank the reviewer for this helpful comment.
>
> **(1) Our focus is on cross-domain knowledge extraction.**
>
> **MoE models are particularly suitable** because routing makes the association between experts and different domains explicit, enabling transferable knowledge to be quantified and extracted.
>
> **(2) In current MoE LLMs, experts are primarily in FFN layers.**
>
> **Most open-source MoE models implement experts in ​FFN blocks**​[1][2][3], where parameters are explicitly modularized. This makes FFN layers the most natural target for extracting reusable expert knowledge.
>
> [1] DeepSeekMoE: Towards Ultimate Expert Specialization in Mixture-of-Experts Language Models
>
> [2] Mixtral of Experts
>
> [3] Qwen3 Technical Report
>
>
> **(3) Attention layers are not explicitly modularized.**
>
> **They are typically shared rather than expertized**, making it difficult to identify domain-agnostic, transferable components in a structured way.
>
> We agree that attention may also encode reusable knowledge, and plan to investigate how to localize and transfer such knowledge in future work.
>
> ***Q3***   **Question3**
>
> ***R3***   Yes, all models are fine-tuned using the same hyperparameters across different baselines. We present a set of hyperparameter settings for the fine-tuning stage as follows:
>
> | Hyperparameter      | Value        |
> | --------------------- | -------------- |
> | Optimizer           | AdamW        |
> | Learning rate       | 3e-5  |
> | Weight decay        | 0.01   |
> | LR scheduler        | Cosine       |
> | Warmup ratio        | 0.03   |
> | Batch size          | 8   |
> | Training epochs （dialogue generation benchmarks）     | 10 |
> | Training epochs （language understanding benchmarks）     | 3 |
> | Max sequence length | 2048  |
>
> Specifically, we keep the SFT setup identical, including learning rate, batch size, number of training steps, and optimization settings, and only vary the initialization strategy. This ensures that the comparison focuses on the effect of the initialization method rather than differences in fine-tuning configurations.
>
> ***Q4***   **Weakness3**
>
> ***R4***   We agree that the choice of Tucker ranks is currently empirical, and a more principled strategy would further strengthen the framework. In our current experiments, the rank is selected from a moderate range to balance two factors: **(i) preserving the dominant shared structure across experts**, and **(ii) avoiding over-parameterization that may reintroduce expert-specific noise**. As shown in Table 4, performance is **relatively stable within a reasonable range of ranks**, suggesting that the method is not overly sensitive to precise tuning. We agree that a more principled approach is desirable. In particular, one promising direction is to determine the rank based on the **energy of the decomposed components** (e.g., retaining a fixed proportion of variance), or to adapt the rank according to model scale and parameter budget. We will add discussion along these lines in the revision.

---

> > ### Author Rebuttal · Reviewer_MWRy · 2026-04-04
> >
> > I thank the authors for their detailed rebuttal and for providing the additional experimental results.
> >
> > 1. The new experimental results are highly appreciated. They effectively address my concerns regarding the positioning of XPERT against other distillation and pruning methods. I look forward to seeing the more principled discussion on Tucker rank selection in the revised manuscript.
> >
> > 2. I acknowledge the authors' rationale for focusing on FFN layers. Given that current MoE architectures explicitly modularize knowledge within FFN blocks, targeting these for expert knowledge extraction is a sound and practical choice.
> >
> > 3. I have also read the response to Reviewer #wews regarding the distinction between "knowledge extraction" and "model compression." While I still consider the boundary between these two concepts to be somewhat blurred, I believe this conceptual nuance does not detract from the overall technical contribution and empirical value of the paper.
> >
> > Overall, the authors have successfully addressed my questions. I am increasing my score to 5 (Accept).

---

> > > ### Author Response · Authors · 2026-04-04
> > >
> > > Sincere thanks for the reviewer's encouraging comments and for increasing the evaluation score of our manuscript. We are very grateful for the reviewer’s recognition of our efforts in revising and improving the paper. We wish the reviewer continued success in their research.

---

### Official Review · Reviewer_N2z3 · 2026-03-01

**Soundness:** 1
**Presentation:** 3
**Significance:** 2
**Originality:** 3
**Overall Recommendation:** 2
**Confidence:** 4

**Summary:**

This paper introduces a method (XPERT) of initializing the feedforward of a Transformer with a matrix derived from the FFN experts of a trained MoE Transformer model. The procedure consists of several steps: 1) measure activation of experts on a variety of domains, then, averaging across the domains, find the experts with the highest |mean - mean absolute deviation| of activation probability, in other words, the experts that are activated often and consistently across domains, 2) stack the weight matrix of each chosen expert, then apply Tucker decomposition to the resulting 3-D tensor to compress the stacked tensor into a core and 3 factor matrices, which, if multiplied together, form an approximation of the original stacked tensor, 3) modify the 3rd factor matrix, corresponding to the mode representing experts, averaging across experts to result in a 1-dimensional vector, then do the "decompression" multiplication described above, resulting in a single weight matrix, 4) reshape the resulting weight matrix so that it can be used to initialize any size target FFN by performing truncated SVD to score rows and columns, then retain only the highest scored rows and columns, duplicating where needed.
Results comparing pretraining Transformers from XPERT-initialized FFNS to pretraining from random initialization and distillation methods. Evaluations are performed after additional finetuning on downstream tasks such as Hellaswag and PIQA and on dialogue tasks.

**Compliance With Llm Reviewing Policy:**

Affirmed.

**Key Questions For Authors:**

1. Why does this work present so many near-random results, and omit all CE Loss or Perplexity results? Do you have results for additional tasks?
2. Can you explain the lack of well ordered results by size for both baselines and your method?
3. What is the rationale for using Tucker Decomposition as opposed to other methods for decomposition and choosing significant tensor subspace, aside from the performance arguments? What are possible alternatives? And did you try the baseline I describe above, which compresses a dense model's FFNs instead?
4. Do you have more thorough hyperparameter sweeps which show the inflection point, at which the lr is too high for all methods? And can you clarify the problem of reported learning rates not matching?
5. Related to the dense baseline, why is this method not extended to non-FFN parameters? Is it not possible to apply the step 4) reshaping to the self attention weight matrices as well and also initialize from those matrices?

**Limitations:**

yes to societal impacts, see remainder of review for scientific limitations

**Strengths And Weaknesses:**

Strengths:
- upcycling existing models is an understudied area, and the field is yet to see a consistent which yields strong, consistent performance gains
- the proposed method is relatively simple and is not computationally expensive. It's presented clearly.
- Some dialogue generation task results are promising
- Authors provide some interesting ablations

Weaknesses:
- Limited model scales, both on the smaller and the larger end, as all models are 270-570M parameters, and all are trained with 5B data tokens (2B & 4B in the appendix, but these may simply have been the same run as 5B, stopped partway through the cosine lr schedule, rather than separate runs with properly annealed checkpoints)
- Tasks chosen for pretrained models are largely unsuitable for these model-data scales, and this is obvious in the performance: hellaswag has a random baseline of 0.25, winogrande of 0.5, piqa of 0.5. Most results for these tasks are so close to random baseline (within 5%, often within 1-2%) as to be difficult to form conclusions from
- Task choice is generally somewhat confusing: Several dialogue tasks chosen are not especially common. There is no reporting of pretraining train and evaluation perplexity / cross-entropy loss, which is much more interpretable at these scales, outside of a single snapshot for one task and one model in Figure 4
- Some results seem nonsensical: For almost all of the tasks, the larger models performance comparably, often _worse_ than the smaller models. This doesn't make any sense if combined with the author's claims that their method has a clear performance improvement. Perhaps one could claim that the baseline methods perform at random chance, and therefore all baseline methods' performance on some tasks is uninterpretable, but this doesn't explain the fact that the XPERT model performances are also not well ordered by size.
- The hyperparameter sweep referenced in the appendix only considers 1, 3, 5 e-5, and of these, 5e-5 yields by far the strongest performance for both the scratch baselines and XPERT-OLMoE, raising the question of why the authors didn't consider even larger LRs to ensure a more fair comparison. Additionally, this doesn't match with the learning rates provided in Appendix B, which are an order of magnitude larger.
- the work is missing an obvious baseline -- rather than choosing the best approximation of all top experts from an MoE model, simply take the FFN from a dense model and compress it to the target FFN size with the author's methodology in step 4).

---

> ### Author Rebuttal · Authors · 2026-03-30
>
> ****Response to Reviewer #N2z3****
> Thank you for your careful reading and insightful questions.
>
> ***Q1:*** **Q1 & W1 & W2 & W3**
>
> ***R1***
>
> **(1) Response to Weakness 1: limited model and data scale.**
> This choice is intentional: our goal is to ​**test whether expert knowledge extracted from source MoE models is reusable**​, and smaller target models provide a controlled setting for isolating initialization effects. **Our current computational resources also limit the scale of both the target models and pre-training data.** We agree that larger-scale validation is important and would be a valuable future direction.
>
> **(2) near-random absolute scores on some tasks.**
> We agree that some benchmarks remain challenging at the current scale, which leads to relatively low absolute scores. However, this is not specific to XPERT. Under the same model and training budget, XPERT shows **clear and consistent gains** over Scratch. For example, the 480M XPERT model improves over Scratch by **2.20** points on **BoolQ** and **3.38** points on ​**PIQA**​, and outperforms Scratch across all evaluated datasets. In addition, on tasks such as **BoolQ** and ​**CaseHold**​, the models achieve clearly non-trivial absolute performance.
>
> **(3) missing CE loss / perplexity.**
> During pre-training, XPERT consistently achieves **lower loss and perplexity** and **faster convergence** than random initialization. Due to space and format limitations during the discussion stage, these results were not included in the current manuscript, but we will add the corresponding curves in the revision.
>
> **(4) task choice and additional tasks.**
> In the current submission, we already evaluate on benchmarks spanning both understanding and generation (e.g., ​**MMLU, S-NI, DollyEval**​), and we will include additional results in the revision to make the evaluation more comprehensive.
>
> ***Q2:*** **Q2 & W4**
>
> ***R2*** We agree that, in general, larger models are expected to perform better. However, the lack of strictly monotonic ordering in our results mainly reflects the **small-scale experimental regime** considered in this work. Prior work has shown that under non-compute-optimal settings, smaller but better-matched models can outperform larger ones [1].
>
> Importantly, our goal is **not** to study scaling laws, but to evaluate the effect of initialization under controlled settings. In this context, the relevant comparison is ​**within each model scale**​.
>
> [1] Training Compute-Optimal Large Language Models, NeurlIPS 2022
>
> ***Q3:*** **Q3 & W6**
>
> ***R3***
>
> **(1) Tucker is chosen for structural reasons.**
> We stack multiple selected experts into a tensor whose modes correspond to ​**input dimension, output dimension, and expert index**​. Tucker decomposition is well suited to this setting because it explicitly extracts a ​**shared low-dimensional subspace across experts**​, which matches our goal of isolating ​**cross-domain, generalizable knowledge**​.
>
> Methods such as **averaging** or **SVD** operate on individual matrices and do not explicitly model the multi-expert structure, while **CP decomposition** imposes a more restrictive factorization. This is also consistent with our ablation results in ​**Table 3**​.
>
> **(2) We have added the suggested dense-FFN baseline.**
> Following the reviewer’s suggestion, we apply the same decomposition and resizing procedure to the ​**FFN weights of a dense model (OLMo-7B)**​. The resulting baseline performs worse than XPERT:
>
> |Method|BoolQ|PIQA|Casehold|Dolly|UnNI|
> |-|-|-|-|-|-|
> |Dense-FFN|72.29|52.94|81.35|22.87|20.07|
> |XPERT|**73.36**|**55.28**|**82.90**|**23.36**|**21.83**|
>
> In dense-FFN, knowledge is more diffusely entangled across parameters and mixes both domain-agnostic and domain-specific information, making reusable cross-domain components harder to isolate. By contrast, MoE routing enables us to identify experts that are consistently activated across domains, allowing XPERT to extract more precise and transferable expert knowledge.
>
> ***Q4:*** **Q4 & W5**
>
> ***R4***
>
> **(1) We have already provided a learning-rate sensitivity analysis in the appendix.**
>
> As shown in Table 8, we evaluate **1e-5, 3e-5, and 5e-5** on representative downstream tasks. Performance improves from **1e-5** to ​**3e-5**​, but already degrades at **5e-5** for both Scratch and XPERT-based methods. This suggests that **3e-5** is already close to the optimal region, and that the turning point is visible within the tested range.
>
> **(2) We keep the same learning-rate sweep across all methods to ensure fairness.**
>
> **(3) Typographical error correction**
>
> **We thank the reviewer for pointing out the inconsistency in the reported learning rates**. The fine-tuning **learning rates listed in Appendix B contain a typographical error**.  The actual learning rate during fine-tuning stage is **3e-5**.
>
> ***Q5:*** **Q5**
>
> ***R5***
> **We sincerely suggest that the reviewers refer to our response (R2) to Reviewer #LZ8p, as his questions are similar to yours.**

---

> > ### Author Rebuttal · Reviewer_N2z3 · 2026-04-04
> >
> > I appreciate the authors' responses. My questions around Q3 are largely resolved or at least sufficiently alleviated, and I appreciate the additional baseline. From Q4, there is some possibility that I misread a table in my initial readings, and set my concerns aside for now. For Q5, I considered this more of a curiosity than a serious concern, and would have liked to see these results but don't consider them core to this paper.
> > However, my main concerns around Q1 and Q2 are not resolved and can not easily be resolved without additional results. I'm not concerned by the _max_ model scale, as the author's seem to believe, but by the limited range of scales, spanning only about a multiple of 2, rather than 1 or 2 orders of magnitude. The problem can be partially addressed by scaling down as well. The current limited scale range renders very challenging any kind of reasoning about larger or smaller scales. The choice of data budget is also odd, at only about 10 times more data than parameters for the largest model scale. I'm also concerned by the noisiness of results. About a quarter to a third of results defy the expected trends, raising serious questions about the trustworthiness of the results. Though it's true that 1 or 2 tasks are relatively more consistent, in the absence of other downstream results, they do not provide sufficiently strong evidence and are somewhat undermined by the noisiness of other results.

---

> > > ### Author Response · Authors · 2026-04-05
> > >
> > > We sincerely thank the reviewer for the detailed follow-up and for clarifying the remaining concerns. We understand that the main issues now center on **(Q1)** the limited scale range and data budget, and **(Q2)** the noisiness and apparent inconsistency of downstream results.
> > >
> > > **Q1: Limited scale range and data budget**
> > >
> > > We agree that the current range of target model sizes (≈2×) is limited and does not support reasoning over one or two orders of magnitude. Our intent is **not** to derive or validate scaling laws, nor to extrapolate behavior to significantly larger or smaller regimes. Rather, our goal is to evaluate whether expert knowledge from MoE models can serve as an effective ​**initialization prior under controlled settings**​, and we will revise the paper to clearly restrict our claims to this scope.
> > >
> > > Regarding the data budget, we acknowledge that the current token-to-parameter ratio is relatively modest. This choice is primarily due to computational constraints and the need for controlled comparisons. We will clarify this limitation and avoid overgeneralizing beyond this regime.
> > >
> > > **Q2: Noisy results and lack of clear trends**
> > >
> > > Importantly, our main claim is ​**not about cross-scale ordering**​, but about ​**within-scale comparison**​: under the same architecture, data, and optimization setting, XPERT consistently improves training dynamics relative to Scratch. This is supported by (please refer to our reply to Reviewer#wews):
> > >
> > > * the newly added ​**pre-training loss/perplexity curves**​, which show consistently faster convergence;
> > > * additional **Scratch+LoRA vs. XPERT+LoRA** results, where XPERT demonstrates both faster SFT convergence and better final performance under identical conditions.
> > >
> > > These complementary results provide a more stable and consistent signal, suggesting that the observed gains are not solely due to noise in individual downstream tasks.
> > >
> > > We sincerely thank the reviewer again for the constructive feedback.

---

### Official Review · Reviewer_LZ8p · 2026-03-09

**Soundness:** 3
**Presentation:** 4
**Significance:** 3
**Originality:** 3
**Overall Recommendation:** 4
**Confidence:** 3

**Summary:**

This paper proposes XPERT, a framework for selecting the experts in MoE withcommon knowledge and reusing the weights of these selections from pre-trained model ti improve training of smaller dense models. XPERT operates in three steps: 1. Expert with common knowledge are selected. 2. Knowledge consolidation are made to to transfer these knowledge into a compact new model. 3. Adapt the dimension of these parameters to the target model to initialize the FFN layers. Experiments are conducted with 270-570 million model size and Llama architecture, and show improvement over traning from both scratch and knowledge distillation, with up to 5 times efficiency gains.

**Compliance With Llm Reviewing Policy:**

Affirmed.

**Final Justification:**

The replies R2 and R4 are solid complements to the original work. R2 directly addresses the MoE-specific design choice, and R4 demonstrates consistent gains at a larger data budget.
For R3, while the 1B teacher experiment is a reasonable addition, the BoolQ result (71.16 vs. 72.66 for the 7B teacher) raises a question: is this drop consistent with the general expectation that larger teachers yield better distillation performance, or is it an uncontrolled fluctuation? This warrants clearer discussion.
R1 does not address my original concern about model scale.
I maintain my score

**Key Questions For Authors:**

1. Does the performance gain still happen with large scall model size?
2. Distill a 270M student with 7B teacher is a degreded way for distillation study due to the large capacity mismatch.
3. How about the results if the pretrain tokens are increased?

**Limitations:**

Yes

**Strengths And Weaknesses:**

Strengths
- This work proposed a well-motivated and straithforward way the transfer the conmmon knowledge into samller model. The method is clearly explained.
- The pipeline is computationally efficient even compared with the distillation alternatives
- Ablation study are well-structured. Each of the three steps are individually ablated.

Weaknesses:
- Target model size is small with only 270M-570M model weights, even compared with the size of  delivered "small" model in current trend.
- The method only initializes FFN layers, leaving attention parameters randomly initialized. No analysis is provided on whether cross-domain knowledge also exists in attention heads and whether transferring those would provide further gains
- pre-training data budgets is only 10B tokens at most, which is also quite small

---

> ### Author Rebuttal · Authors · 2026-03-30
>
> ****Response to Reviewer #LZ8p****
> Thank you for your careful reading and insightful questions.
>
> ***Q1***   **Weakness1 & Question1**
>
> ***R1***   We thank the reviewer for this observation. We acknowledge that the target models considered in this work (270M–570M) are smaller than many contemporary large-scale models.
>
> **(1) Our goal is to validate expert knowledge reuse.**
>
> This work focuses on **whether expert knowledge extracted from MoE models can improve training**. Smaller target models provide a more controlled setting to isolate and evaluate this effect.
>
> **(2) Larger-scale experiments are limited by current resources.**
>
> We currently do not have sufficient resources to train significantly larger models under the same setup. We agree that evaluating at larger scales is important and will clarify this limitation in the paper.
>
> We emphasize that our method is not tied to a specific model size. **We view this work as a step toward understanding expert knowledge reuse, and scaling to larger models is a natural next step.**
>
> ***Q2***   **Weakness2**
>
> ***R2***    We thank the reviewer for this helpful comment.
>
> **(1) Our focus is on cross-domain knowledge extraction.**
>
> **MoE models are particularly suitable** because routing makes the association between experts and different domains explicit, enabling transferable knowledge to be quantified and extracted.
>
> **(2) In current MoE LLMs, experts are primarily in FFN layers.**
>
> **Most open-source MoE models implement experts in ​FFN blocks**​[1][2][3], where parameters are explicitly modularized. This makes FFN layers the most natural target for extracting reusable expert knowledge.
>
> **(3) Attention layers are not explicitly modularized.**
>
> **They are typically shared rather than expertized**, making it difficult to identify domain-agnostic, transferable components in a structured way.
>
> We agree that attention may also encode reusable knowledge, and plan to investigate how to localize and transfer such knowledge in future work.
>
> [1] DeepSeekMoE: Towards Ultimate Expert Specialization in Mixture-of-Experts Language Models
>
> [2] Mixtral of Experts
>
> [3] Qwen3 Technical Report
>
> ***Q3***   **Question2**
>
> ***R3***   We thank the reviewer for this important observation.
>
> **(1) Our original 7B-teacher distillation baseline was chosen for fairness.**
>
> We use a **7B teacher model** in the original distillation baseline because XPERT also uses a ​**7B source model**​, and we would like to keep the source model scale consistent across methods. This ensures that the comparison is made under the same source-model setting, rather than confounded by differences in teacher/source capacity.
>
> **(2) We have added a smaller-teacher distillation baseline.**
>
> To address this concern, we include an additional experiment using a **1B teacher model (Llama-3.2-1B)** for distillation, which reduces the capacity mismatch compared to the original 7B→270M setting.
>
> | Baseline | Model Size | BoolQ | PIQA | DollyEval | UnNI |
> | --- | --- | --- | --- | --- | --- |
> | Distillation (1B teacher) | 270M | 71.16 | 51.74 | 23.30 | 21.14 |
> | Distillation (7B teacher) | 270M | 72.66 | 52.39 | 22.43 | 20.50 |
> | XPERT | 270M | **73.24** | **54.46** | **23.67** | **21.55** |
>
> XPERT consistently outperforms distillation under both teacher sizes.
>
> **(3) This suggests the advantage is not due to teacher size mismatch.**
> Even when the teacher–student gap is reduced, distillation remains less effective, indicating that the limitation lies in the ​**behavior-level transfer mechanism**​, whereas XPERT transfers ​**structured expert knowledge at the parameter level**​, leading to more effective reuse.
>
> ***Q4***   **Weakness3 & Question3**
>
> ***R4*** We thank the reviewer for this question.
>
> **(1) We have added a larger-budget experiment.**
>
> To address this concern, we conducted an additional experiment with ​**20B pre-training tokens**​. The advantage of XPERT remains clear at this larger budget: it still converges faster and achieves better downstream performance than random initialization under the same setup.
>
> | Baseline/Dataset | Model SIze | DollyEval | S-NI | UnNI |
> | --- | --- | --- |  --- |  --- |
> | Scratch | 570M | 20.63 | 12.32 | 16.06 |
> | XPERT | 570M | **23.38** | **16.91** | **22.17** |
>
> **(2) This suggests that the benefit of XPERT is not limited to the smallest pre-training budgets.**
>
> **(3) We agree that larger-scale verification would be valuable.**
>
> **However, due to current compute constraints**, we are not able to run substantially larger pre-training budgets in this submission. We will clarify this limitation in the paper and view larger-scale evaluation as an important direction for future work.

---

> > ### Author Rebuttal · Reviewer_LZ8p · 2026-04-07
> >
> > The replies R2 and R4 are solid complements to the original work. R2 directly addresses the MoE-specific design choice, and R4 demonstrates consistent gains at a larger data budget.
> > For R3, while the 1B teacher experiment is a reasonable addition, the BoolQ result (71.16 vs. 72.66 for the 7B teacher) raises a question: is this drop consistent with the general expectation that larger teachers yield better distillation performance, or is it an uncontrolled fluctuation? This warrants clearer discussion.
> > R1 does not address my original concern about model scale.
> > I maintain my score

---

> > > ### Author Response · Authors · 2026-04-07
> > >
> > > We thank the reviewer for the detailed follow-up and for raising important concerns regarding both the scale range and the interpretation of the distillation results.
> > >
> > > **(1) On the scale range and data budget.**
> > >
> > > We agree that the current range of model sizes is limited. We acknowledge that broader coverage over model scales and data budgets would strengthen the conclusions, and we will explicitly narrow the corresponding claims and discuss this limitation in the revised version. **Due to computational constraints during the rebuttal phase**, extending the experiments to a wider range of scales is not feasible at this stage, **but we consider it an important direction for future work.**
> > >
> > > **(2) On the distillation results.**
> > >
> > > We thank the reviewer for this question. We agree that the relationship between teacher scale and distillation performance is not necessarily monotonic in our setting. While a larger teacher may provide a stronger supervision signal, it also introduces a larger teacher–student capacity gap, which can make knowledge transfer less well matched to the student.
> > >
> > > Therefore, we do not interpret the difference between the 1B and 7B teachers as evidence of a simple monotonic trend. Rather, we view it as indicating that **teacher scale and teacher–student mismatch may both affect distillation quality.** In our results, the 7B teacher gives slightly better performance on some tasks (e.g., BoolQ, PIQA), while differences on other tasks remain small. We will clarify this point in the revision and avoid over-interpreting the teacher-scale comparison.
> > >
> > > We sincerely thank the reviewer again for the thoughtful questions and constructive feedback, which have been very helpful in improving the clarity and presentation of our work.

---

### Official Review · Reviewer_wews · 2026-03-13

**Soundness:** 2
**Presentation:** 3
**Significance:** 2
**Originality:** 3
**Overall Recommendation:** 4
**Confidence:** 3

**Summary:**

This work proposes an approach for training models from partially initialized weights. The partial initialization is performed by transplanting a compressed MoE layer from another pretrained model. Experiments show improvements over training from scratch on OLMo- and LLaMA-like architectures.

**Compliance With Llm Reviewing Policy:**

Affirmed.

**Final Justification:**

I thank the authors for their rebuttal and the additional experiments. Providing the exact XPERT + LoRA vs. Scratch + LoRA comparison directly resolves my primary concern (W3) by proving that XPERT offers a strictly better initialization prior than the baseline. Furthermore, the new ablations on noisy replication and layer selection (W2), along with the clarifications on pre-training dynamics (Q1), adequately address my remaining technical questions. While the overall experimental scale remains somewhat limited, the updated empirical evidence now concretely supports the paper's core claim: MoE expert knowledge can be effectively extracted to improve the initialization and training of dense models. Because the proposed method is novel, intuitive, and computationally efficient, the paper's strengths outweigh its limitations. I am raising my score to a Weak Accept.

**Key Questions For Authors:**

When you initialize only the FFN blocks with consolidated pre-trained expert weights but leave Multi-Head Attention randomly initialized, what evidence shows the transferred FFN structure is not rapidly overwritten in early training?

**Limitations:**

yes

**Strengths And Weaknesses:**

## Strengths

- While traditional model compression methods such as knowledge distillation or pruning typically operate in dense to dense or MoE to MoE settings, transferring knowledge from a sparse MoE model into a dense model is relatively unexplored and represents a novel direction.
- The paper’s narrative of extracting, consolidating, and reusing expert knowledge is intuitive and easy to follow. The authors do a good job framing the method and clearly explaining the what and how of the XPERT pipeline.
- The paper clearly delineates the scope of the experimental setup and the specific model families it targets.

## Weaknesses

- Although the MoE to dense transfer setting is novel, the work appears to rely on post training compression techniques as a precursor to training a new model further, which is not well motivated and feels conceptually misaligned.
- The presentation lacks sufficient justification for key design choices. Important architectural and methodological decisions such as choosing exact replication over noisy replication, or selecting the first N layers rather than sampling layers uniformly, are stated without supporting ablations or theoretical rationale.
- From a practical perspective, it is unclear why this particular partial initialization strategy, specifically using a compressed MoE transplant, is preferable to simpler baselines such as initializing from an arbitrary subset of the donor model’s weights or applying standard LoRA fine tuning, which are more straightforward and may yield more reliable results.

---

> ### Author Rebuttal · Authors · 2026-03-30
>
> ****Response to Reviewer #wews****
> Thank you for your careful reading and insightful questions.
>
> ***Q1*** **Weakness1**
>
> ***R1***  We would like to clarify that our method is **not** intended as a post-training compression pipeline.
>
> **(1) Our goal is knowledge reuse rather than source-model approximation.**
>
> Unlike pruning or distillation, our objective is not to produce a compressed surrogate of the source model, but to extract **cross-domain, reusable expert knowledge** and transfer it to improve the training of new models. The extracted representation is therefore evaluated by its effect on ​**optimization, convergence, and downstream performance**​, rather than by reconstruction fidelity.
>
> **(2) The decomposition step serves knowledge extraction, not compression.**
>
> It is used to identify and consolidate ​**shared functional subspaces across experts**​, especially the generalizable structure recurring across domains, rather than to reduce the source model itself.
>
> **(3) XPERT also differs from distillation in practical usage.**
>
> It performs a **one-time extraction** from the source model, after which the extracted knowledge can be adapted to target models of different sizes and reused flexibly, without repeatedly invoking the source model during training.
>
> To further clarify the practical difference from distillation-based approaches, we compare the **pre-training cost** under the same setting (5B tokens, 390M model, NVIDIA H100):
>
> |Method|Size|GPU*h (H100)|
> |-|-|-|
> |Scratch|390M|2*41|
> |Distillation|390M|2*320|
> |XPERT|390M|2*41|
>
> XPERT has essentially the ​**same training cost as Scratch**​, while distillation is **significantly more expensive** due to repeated teacher forward passes during training. This further supports that XPERT should be viewed as a **knowledge reuse framework** rather than a compression-based or teacher-dependent training method.
>
> ***Q2*** **Weakness2**
>
> ***R2*** To better justify these design choices, we add ablations on both replication strategy and layer selection.
>
> **(1) Exact replication vs. noisy replication.**
>
> Introducing noise consistently degrades performance:
>
> |Method (570M)|PIQA|BoolQ|Casehold|Dolly|
> |-|-|-|-|-|
> |Exact Replication|55.44|73.91|82.96|24.19|
> |Noisy replication (1e-5)|52.45|72.02|81.48|20.64|
> |Noisy replication (1e-4)|51.85|71.90|81.48|19.91|
>
> This suggests that the extracted expert parameters contain **structured knowledge** that is sensitive to perturbation, so exact replication is important for preserving their integrity.
>
> **(2) Layer selection strategy.**
>
> We further compare selecting the **first N layers** with uniform layer sampling:
>
> |Layer Selection Strategy (8 Layers)|BoolQ|PIQA|Casehold|AVG|
> |-|-|-|-|-|
> |Uniform Selection|70.70|52.07|81.35|68.04|
> |First 8 layers|**72.10**|**52.29**|**82.96**|**69.12**|
>
> Selecting the first layers consistently outperforms uniform sampling. We attribute this to the fact that ​**earlier layers tend to capture more general and transferable representations**​, while higher layers are more task- or domain-specific. Therefore, prioritizing lower layers better aligns with our goal of extracting ​**cross-domain reusable knowledge**​.
>
> ***Q3*** **Weakness3**
>
> ***R3***
>
> **(1) XPERT is not arbitrary partial initialization.**
>
> Our method does not copy an arbitrary subset of donor weights. Instead, it identifies experts that are consistently activated across diverse domains and consolidates them to extract ​**shared, generalizable structure**​. This provides a structured inductive bias, rather than unstructured parameter copying, and explains why XPERT improves over random initialization.
>
> **(2) XPERT and LoRA are complementary, not competing.**
>
> LoRA is a **parameter-efficient fine-tuning method** that learns low-rank updates from downstream data. XPERT instead provides a ​**knowledge-informed initialization prior to training**​, derived from a source MoE model. Put simply, XPERT improves ​**where training starts**​, while LoRA improves ​**how parameters are updated during training**​. These two ideas are therefore compatible rather than mutually exclusive.
>
> ***Q4*** **Question**
>
> ***R4***    Our empirical results suggest that it is ​**not rapidly overwritten**​. XPERT-initialized models show **faster convergence and better early-stage performance** than random initialization, and this advantage persists throughout fine-tuning, as shown in ​**Figure 4**​. If the initialized FFN structure were quickly destroyed, such a consistent gap would be unlikely to persist.
>
> We observe the same pattern during ​**pre-training**​: XPERT yields consistently **lower loss and perplexity** and reaches convergence faster than random initialization. This indicates that the effect of initialization is sustained throughout training rather than being limited to the earliest steps.
>
> **Due to rebuttal format limitations, we cannot update the manuscript at this stage, but we will include the pre-training loss and perplexity curves in the revision**

---

> > ### Author Rebuttal · Reviewer_wews · 2026-04-04
> >
> > I thank the authors for their detailed rebuttal and for providing additional experimental results. My concerns regarding **W1**, **W2**, and **Q1** are now largely resolved, or at least sufficiently alleviated. I appreciate the additional comparisons and results provided both here and in response to the other reviewers. I found the results of this work quite interesting, and it certainly has merit.
> >
> > However, my main concern regarding **W3** has not been resolved and cannot easily be addressed without additional results. I understand that the authors are limited in time and compute, but in order to be convinced, I would like to request ONE experiment from the following two options:
> >
> > 1. A comparison of training curves for XPERT vs. the baseline, where all layers are taken from the pre-trained student except the FFN, which is randomly initialized, under the same optimization setting, OR
> > 2. A comparison of training curves for LoRA on a scratch student vs. LoRA on an XPERT-initialized student, under the same optimization setting (just training, not distillation).
> >
> > Either of these could help convince me that the knowledge transfer is indeed sufficient. This could be done at a scale suitable for the authors, such as **1B** parameters (or 7B, if possible). *If it is possible to provide such a result, I would raise my score.*
> >
> > **Remark**
> >
> > I found it quite interesting that “earlier layers tend to capture more general and transferable representations.” Have you conducted a more fine-grained analysis of which layers are more important? Usually, it seems that the first two-thirds of the layers are more active in terms of knowledge contextualization. The answer to this remark would not affect my final score; it is simply a question out of reviewers' curiosity.

---

> > > ### Author Response · Authors · 2026-04-05
> > >
> > > We sincerely thank the reviewer for the insightful suggestion and for the positive assessment of our work. We greatly appreciate the reviewer’s effort in helping us further improve the clarity and completeness of the paper.
> > >
> > > Following the reviewer’s suggestion, we provide additional experiments to better justify the effectiveness of XPERT:
> > >
> > > **(1) Pre-training dynamics.**
> > > We add training curves comparing XPERT and Scratch initialization during pre-training, including both loss and perplexity. **These results show that XPERT consistently leads to faster convergence.**
> > >
> > > **(2) LoRA-based comparison (addressing W3).**
> > > To directly address the reviewer’s concern regarding simpler baselines such as LoRA, we further provide training curves comparing:
> > >
> > > * LoRA on a randomly initialized model (Scratch + LoRA)
> > > * LoRA on an XPERT-initialized model (XPERT + LoRA)
> > >   under identical optimization settings.
> > >
> > > The curves include both training loss and evaluation metrics (e.g., Accuracy / ROUGE-L) across multiple datasets. **All results are available at the anonymous repository:** https://anonymous.4open.science/r/response_to_Reviewer-wews-757D/​
> > >
> > > In addition to the training curves, we include a table summarizing the **SFT performance** of **Scratch + LoRA** and **XPERT + LoRA** across datasets:
> > >
> > > |Method/Dataset|BoolQ|PIQA|Dolly|UnNI|
> > > |-|-|-|-|-|
> > > |Scratch + LoRA|67.15|50.76|20.86|20.57|
> > > |XPERT + LoRA|**70.52**|**51.20**|**22.82**|**24.31**|
> > >
> > > **Findings.**
> > >
> > > These additional results show that:
> > >
> > > * XPERT initialization leads to **faster convergence** during both pre-training and SFT;
> > > * XPERT remains effective even under parameter-efficient training (LoRA), demonstrating that it is ​**complementary to simpler approaches rather than redundant**​.
> > >
> > > We also observe that LoRA-based training yields lower absolute performance compared to full-parameter fine-tuning, likely due to limited model capacity. However, the relative advantage of XPERT remains consistent.
> > >
> > > **Due to time and computational constraints during the rebuttal phase**, all additional experiments are conducted on the largest target model scale used in the paper (570M parameters). We acknowledge that evaluating XPERT at larger scales (e.g., 3B or 7B) would be valuable, and we plan to explore this direction in future work when resources permit.
> > >
> > > ***Additional discussion on layer-wise effects***
> > >
> > > We thank the reviewer for this interesting observation. While we do not perform a fully fine-grained layer-wise analysis, our results in the Appendix show that knowledge extracted from earlier layers leads to a little better transfer performance than that from later layers.
> > >
> > > We also find the reviewer’s intuition regarding the first two-thirds of layers insightful. A possible explanation is that earlier and middle layers tend to encode **more general-purpose and contextual representations**, whereas later layers become increasingly task- or output-specific. This makes knowledge from earlier layers more transferable across domains, which is consistent with our observations. A more detailed layer-wise analysis would be an interesting direction for future work.
> > >
> > > **We sincerely thank the reviewer again for the valuable feedback and constructive suggestions. We will incorporate the additional analysis and experimental results into the revised version of the paper.**

---

### Decision · Program_Chairs · 2026-04-30

**Decision:**

Accept (regular)

**Comment:**

This paper proposes XPERT, a framework for selecting the experts in MoE with common knowledge and reusing the weights of these selections from a pre-trained model to improve the training of smaller dense models. The framework employs Tucker decomposition for parameter extraction and a mechanism for adaptation.

In summary, three reviewers lean to positive (A, WA, WA — MWRy, wews, LZ8p) but one to negative (R — N2z3).

The strengths:

- The paper is well-motivated. The idea of reusing and transferring knowledge from a sparse MoE model into a dense model is relatively unexplored and represents a novel direction. (wews/ N2z3 / MWRy )
- The paper offers a novel perspective on MoE models as reusable "knowledge sources". (N2z3 / MWRy)
- The proposed method is technically grounded and simple. (N2z3 / MWRy)
- The ablation studies are well-structured. (LZ8p / N2z3)

The major common concerns are:

- Insufficient model and data scales — small model size, limited pre-training data. (LZ8p / N2z3)
- Justification or analysis on initialization FFN layers vs. attention layers (MWRy / LZ8p/ MWRy)
- Missing baselines such as LoRA fine-tuning or model pruning. (wews / MWRy / N2z3 / MWRy)
- Experimental evaluation — near-random performance, missing metrics such as PPL, and uncommon tasks. (N2z3
    - N2z3: Some results seem nonsensical. For almost all of the tasks, the larger models perform comparably, often *worse* than the smaller models.
- More clarification of design choices, such as layer selection and hyperparameters. (wews / N2z3 )

According to the rebuttal acknowledgement by reviewers, three reviewers increased their initial scores, satisfying the authors’ responses and additional experimental results.

Despite Reviewer N2z3’s remaining concern, which is a limited scale range and data budget, the paper would contribute to the community with its motivation and novel perspectives on MoE, while obviously restricting the claim to the limited scope. In this sense, I recommend WA.